# Hallucinations and Key Information Extraction in Medical Texts: A Comprehensive Assessment of Open-Source Large Language Models

Anindya Bijoy Das
*The University of Akron, OH, USA*
adas@uakron.edu

Shibbir Ahmed
*Texas State University, TX, USA*
shibbir@txstate.edu

Shahnewaz Karim Sakib
*University of Tennessee at Chattanooga, TN, USA*
shahnewazkarim-sakib@utc.edu

*Abstract*—**Clinical summarization is crucial in healthcare as it distills complex medical data into digestible information, enhancing patient understanding and care management. Large language models (LLMs) have shown significant potential in automating and improving the accuracy of such summarizations due to their advanced natural language understanding capabilities. These models are particularly applicable in the context of summarizing medical/clinical texts, where precise and concise information transfer is essential. In this paper, we investigate the effectiveness of open-source LLMs in extracting key events from discharge reports, including admission reasons, major in-hospital events, and critical follow-up actions. In addition, we also assess the prevalence of various types of hallucinations in the summaries produced by these models. Detecting hallucinations is vital as it directly influences the reliability of the information, potentially affecting patient care and treatment outcomes. We conduct comprehensive simulations to rigorously evaluate the performance of these models, further probing the accuracy and fidelity of the extracted content in clinical summarization. Our results reveal that while the LLMs (e.g., Qwen2.5 and DeepSeek-v2) perform quite well in capturing admission reasons and hospitalization events, they are generally less consistent when it comes to identifying follow-up recommendations, highlighting broader challenges in leveraging LLMs for comprehensive summarization.**

*Index Terms*—**Medical Text Summarization, Large Language Models, Hallucinations, Key Information Extraction.**

## I. INTRODUCTION

Clinical text summarization [1] is a crucial task in modern healthcare, as it enables efficient extraction of key medical information from lengthy and complex documents such as electronic health records (EHRs), discharge summaries, and radiology reports. The vast amount of unstructured textual data generated in clinical settings poses a significant challenge for healthcare professionals, who must rapidly interpret patient histories, diagnoses, and treatment plans to make informed decisions [2]. Effective summarization of medical texts can enhance clinical workflow efficiency and improve patient outcomes by ensuring that essential information is readily accessible [3]. Additionally, it plays a vital role in medical research, enabling quicker literature reviews and knowledge synthesis [4]. However, traditional summarization techniques often struggle with the specialized language, domain-specific jargon, and contextual nuances of medical texts, highlighting the need for more advanced, AI-driven approaches [5].

Large Language Models (LLMs) have emerged as powerful tools in artificial intelligence and machine learning, demonstrat-ing remarkable capabilities in natural language understanding, medical imaging [6], generation, and contextual reasoning [7]. These models, trained on vast and diverse datasets, can comprehend and generate human-like text, making them particularly well-suited for tasks such as summarization, translation, and question-answering [8]. In the medical domain, LLMs offer significant potential for summarizing clinical texts [9] by capturing key information while preserving critical details necessary for decision-making. Unlike traditional rule-based or statistical summarization methods, LLMs can adapt to complex linguistic structures, recognize medical terminologies, and generate concise yet informative summaries tailored to specific clinical needs. By leveraging contextual embeddings and domain-specific fine-tuning, LLMs can improve the accuracy and relevance of medical text summarization, ultimately helping reduce information overload and improve patient care [7].

Despite their potential, LLMs face notable challenges in summarizing medical texts, which can impact the reliability and accuracy of generated summaries. These challenges include:

**Key Information Extraction:** Extracting clinically relevant events from medical texts is a critical requirement for summarization. However, LLMs may struggle to prioritize essential details, such as symptoms, treatments, and medication changes, especially when faced with lengthy or complex narratives. Without proper mechanisms for identifying and preserving crucial information, summaries may omit significant medical events, leading to incomplete or misleading interpretations.

**Hallucinations:** LLMs are known to generate plausible yet incorrect or non-existent information, a phenomenon referred to as hallucination [10], [11], [12]. In the medical domain, hallucinated details in summary, such as incorrect dosages, fabricated conditions, or misrepresented patient histories, thus can have serious consequences for clinical decision-making. Fig. 1 illustrates an example where the LLM-generated summary introduces unsupported or fabricated clinical information. It includes hallucinated clinical events (e.g., "heart attack is unlikely") and incorrect medication details (e.g., "Metformin to 1000 mg"), which were not present in the original discharge report. This issue underscores the need for domain-specific fine-tuning and external fact-checking to ensure the reliability of medical text summaries [13], [14].

In this paper, we aim to highlight and analyze the challenges of key event extraction and hallucinations in LLM-driven

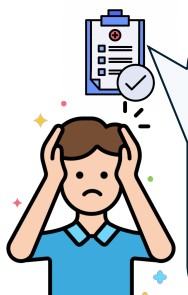

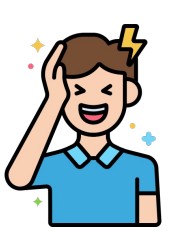

Mrs. ___ is a ___ female w/ PMH of CAD, HTN, HLD, DMII (on Metformin) who presented to the ED with severe dyspnea and chest pain. ____ was admitted for acute on chronic diastolic heart failure in the setting of PNA. ACTIVE ISSUES: # Acute on chronic diastolic heart failure: ECG in ED unremarkable w/out ST-T wave abnormalities, heart enzymes x2 ruled out acute myocardial damage, exacerbation likely in the setting of PNA. Patient reported dietary change w/ increased salt intake that could have caused this. Diuresed with lasix gtt. Balanced negative daily (max. ___ L). Sx improved with diuresis. #Pneumonia: Developed productive cough and new O2 requirement, most likely CAP, CXR showed new LLL infiltrate. Started on IV CTX and azithromycin to complete a 5 day course (held home atorvastatin in the setting of azithromycin). Fever resolved on HD#1. Remained afebrile for entire admission. Significantly improved and was discharged on HD#5. # T2DM: Serum glc slightly elevated on admission. ___ suggested to increase Metformin as outpatient (Dr. _______________). #Hyponatremia: hypervolemic hyponatremia in setting of DHF. Resolved with lasix. # HTN. Normotensive during admission. Continued home lisinopril and metoprolol. TRANSITIONAL ISSUES: Please follow-up with PCP to adjust DM medications.

You were admitted to the Acute Care Service with shortness of breath and chest pain. A heart attack was unlikely, but we performed several tests and ruled out acute damage to your heart. We performed an echocardiography and found a worsening of your heart failure. This was likely due to a pneumonia of your right lung which we found on a chest xray. We treated you with diuretics for your edema and you were started on two antibiotics for pneumonia. You significantly improved during the second day in the hospital. You had a problem with your electrolytes which resolved with diuresis. Your high blood pressure was treated with your home medications. Your high blood pressure was treated with your home medications. We noticed an increased blood sugar up to 220 and we informed your PCP. We suggest increasing your Metformin to 1000 mg. You were discharged to your assisted living facility

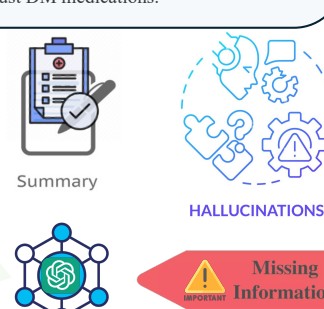

Summary

HALLUCINATIONS

Missing Information

Fig. 1: A sample hospital discharge report (top) where the patient initially could not understand the content. However, by using an LLM, the patient is able to understand the summary (bottom) of the discharge report. Note that the LLM-generated summary may contain hallucinations, and key information may be missing. For instance, the yellow and red highlighted portions are unsupported and incorrect/contradicted fact hallucinations, respectively.

medical text summarization. To achieve this, we conduct a comprehensive evaluation using discharge summaries from the MIMIC-IV dataset [15], a widely used repository of de-identified clinical records. We explore various open-source large language models (LLMs) to assess their effectiveness in extracting essential medical events while minimizing hallucinations. The open-source LLMs used in this study include LLaMA [16], Mistral [17], Gemma [18], Phi [19], Falcon [20], LLaVA [21], DeepSeek [22], and Qwen [23]. Our analysis examines the strengths and limitations of these models in handling domain-specific language and preserving critical clinical information, which is essential for making LLM-driven summarization a practical and trustworthy tool for healthcare applications.

## II. PRIOR WORKS AND OUR CONTRIBUTIONS

Medical text summarization has been an active area of research, with numerous studies exploring techniques to extract essential clinical information while maintaining accuracy and reliability. Traditional NLP approaches have relied on rule-based and machine-learning methods, while recent advancements in deep learning and large language models have significantly improved the capabilities of automatic summarization [24]. However, despite these advancements, two key challenges remain particularly difficult: (1) accurately extracting key medical events and (2) addressing hallucinations in LLM-generated summaries. Addressing these issues is crucial for ensuring that AI-driven summarization tools can be safely and effectively deployed in healthcare settings.

### A. Previous Works

Understanding the challenges of medical text summarization requires examining prior research on key event extraction and hallucination detection in LLM-generated summaries. In the following sections, we discuss existing works addressing these issues and their relevance to our study.

*1) Key Event/Information Extraction:* Extracting essential clinical information from lengthy medical documents has been a long-standing challenge in natural language processing. Traditional approaches relied on rule-based methods and statistical models, such as conditional random fields and hidden Markov models, to identify key events. More recently, deep learning-based techniques, including recurrent neural networks, convolutional neural networks, and transformer-based architectures [25], have been employed to improve accuracy. Pre-trained biomedical models such as BioBERT [26] and ClinicalBERT [27] have demonstrated effectiveness in extracting medical entities from structured and unstructured clinical narratives. However, these methods often require extensive domain-specific training data and struggle with generalization across different medical contexts.

With the advent of LLMs, researchers have explored their potential for key event extraction in clinical text summarization. Models such as GPT-4 [28] and Med-PaLM [29] have shown promise in capturing contextual dependencies, but they may still fail to prioritize clinically relevant insights accurately. Furthermore, without explicit fine-tuning on labeled clinical datasets, LLMs risk omitting critical patient information, making their direct use to medical summarization challenging.

*2) Detection of Hallucination:* A major limitation of LLMs in medical summarization is their tendency to generate hallucinated content, i.e., fabricated or misleading information that does not align with the input text. Hallucinations in medical AI can have serious consequences, potentially leading to incorrect diagnoses, erroneous treatment recommendations, and misinterpretation of patient histories.

Existing research has explored various techniques for hallucination detection and mitigation in NLP applications. Some approaches rely on fact-checking methods that compare generated summaries against source texts using similarity metrics or retrieval-based validation. Others employ uncertainty quantification techniques, such as confidence scoring and probabilistic modeling, to assess the reliability of LLM-generated outputs. In the biomedical domain, external knowledge bases (e.g., SNOMED CT [30]) have been integrated into NLP pipelines to verify the factual accuracy of generated content. However, ensuring faithfulness in LLM-generated medical summaries remains an open challenge, necessitating further research into robust evaluation metrics and hallucination filters.

### B. Summary of Contributions

In this paper, we build upon existing research by conducting a comprehensive analysis of LLM-driven summarization of medical texts, focusing on key event extraction and hallucination detection. Our main contributions include:

- Evaluation of Open-Source LLMs for Clinical Summarization: We explore various publicly available large language models to assess their performance in summarizing discharge summaries from the MIMIC-IV dataset.
- Analysis of Key Event Extraction Capabilities: We investigate how well different LLMs identify and retain critical clinical insights, such as diagnoses, treatments, and medication changes.
- Hallucination Detection and Quantification: We evaluate the extent of hallucinations in LLM-generated summaries and analyze potential factors contributing to misleading or fabricated information.

By systematically examining these challenges, we aim to provide valuable insights into the feasibility of using LLMs for medical text summarization and propose directions for improving their reliability in clinical applications.

### III. ANALYZING LLM-GENERATED CLINICAL SUMMARIES

In this section, we examine the key aspects of essential information extraction and corresponding hallucinations in clinical summarization by LLMs. We illustrate these through a detailed analysis of a representative hospital discharge report, demonstrating both accurate extractions and common errors made by LLMs in real-world clinical settings.

### A. Key Event Extraction in Clinical Summarization

Text summarization is a key task in natural language processing, condensing extensive texts into concise summaries while preserving essential information. It is widely used in domains requiring swift data comprehension, e.g., finance and law, to simplify legal documents while highlighting key trends, risks, facts, and arguments. In medical applications, summarization extracts vital clinical details from documents like patient histories and treatment plans, aiding both healthcare professionals and patients. For example, summarizing a CT scan report can highlight key findings, facilitating swift clinical

decisions. Leveraging open-source LLMs for medical text summarization enhances patient care by improving communication and clarity of critical information.

This study utilizes open-source LLMs to summarize hospital discharge reports, simplifying complex medical jargon for better patient comprehension. We assess their efficacy in generating patient-friendly summaries by extracting key details, including admission reasons, significant hospital events, and essential follow-up actions, analyzed through a specific example. Here, we consider an example of the hospital's discharge report presented in Fig. 2 to systematically extract and assess key clinical data. The methodology identifies admission reasons, key interventions, diagnostic events, and follow-ups, ensuring a clear understanding of the patient's healthcare journey and critical decisions.

In this example, the patient's admission is due to acute symptoms like fever, nausea, and vomiting, along with chronic conditions such as hypertension requiring urgent care. Hospitalization involves administering broad-spectrum antibiotics and conducting extensive diagnostic imaging. The discharge plan includes essential follow-ups, such as liver function tests and a chest CT in six months for ongoing monitoring. We tasked each LLM with generating a summary of this hospital discharge report, limiting their responses to a maximum of 1000 characters. Our goal is to evaluate whether the open-source LLMs effectively capture these critical details in their summaries. Fig. 3 presents some examples of such summaries produced by the LLMs.

Consider the summaries in Fig. 3(a) and 3(b) generated by Gemma2 and LLaMA. While both captured several key events, they overlooked critical details. Notably, neither summary included liver function monitoring or hypertension management in the follow-up plan. Additionally, Gemma2 failed to recognize the patient's existing esophageal cancer diagnosis. Despite being well below the 1000-character limit (505 and 617 characters, respectively), the summaries had enough space to include these essential details but failed to do so.

### B. Hallucination in LLM-Generated Summaries

A major limitation of LLMs in medical applications is their tendency to generate hallucinated content—information that is not present in the source text or is factually incorrect [31]. In the context of clinical summarization, hallucinations can be particularly problematic, as they may introduce inaccurate medical details, misrepresent patient histories, or suggest incorrect treatments, posing risks to patient safety.

Thus, hallucinations in text generated by LLMs can manifest in various forms, including the following:

- (i) Unsupported facts [32]: The model generates clinical details, such as a diagnosis or prescription, that were not present in the original discharge summary. These can include unsupported conditions, procedures, medications, numbers, names and other details.
- (ii) Incorrect or Contradicted Facts [32]: The model misinterprets, distorts, or contradicts events, resulting

Fig. 2: A sample hospital discharge report (from MIMIC-IV dataset). We highlighted some primary reasons for the patient's admission, which may include fever and nausea, some important events during the hospital stay, which may include head CT or oral labetalol, and some essential follow-up actions, which may include chest CT and liver function tests (LFT).

Fig. 3: Discharge report summary generated by (a) Gemma2, (b) LLaMA, (c) Qwen2.5 and (d) Falcon. Some instances of hallucinations are highlighted.

in misleading conclusions that deviate from the original source and potentially alter the intended clinical meaning.

- (iii) Faithfulness hallucinations [33]: The model misses or omits key details of the source, altering the meaning of the summary and affecting its reliability.
- (iv) Content hallucinations [34]: The model introduces content that is completely unrelated or irrelevant to the source document.

In short, while some abstraction is expected in summarization, hallucinations introduce unverifiable or incorrect content. Using the discharge report in Fig. 2, we illustrate this with summaries from (i) Qwen2.5 and (ii) Falcon in Figs. 3(c)-3(d), respectively.

Firstly, the Qwen2.5 summary incorrectly states the patient is 45 years old: an unsupported fact not found in the report. It also omits essential follow-up instructions for Liver Function Tests (LFTs), despite the report explicitly stating their necessity twice. This is particularly concerning given that timely LFT monitoring can be crucial for managing the patient's complications noted during hospitalization. With 122 characters still available, this omission reflects a faithfulness hallucination, where the model fails to include essential care details.

In contrast, Falcon's summary inaccurately states that a biopsy "is recommended", whereas the original report confirms it was already rescheduled, misrepresenting the planned action. It also falsely claims the patient received antibiotics for 24 hours, while the report states they were monitored without

antibiotics for over 24 hours, both errors falling under "incorrect facts hallucinations". Additionally, the summary inconsistently switches between 'they' and 'he' (e.g., "They were advised to limit alcohol..." vs. "He is also recommended for a repeat LFT".), causing gender hallucinations that compromise clarity.

Our evaluation method, focused on extracting key information and spotting mistakes, provides useful insights for creating better discharge summaries. These results can help clinical teams improve language model tools that support usability, readability, and patient comprehension in real-world healthcare settings by by ensuring accurate and clear communication.

## IV. EXPERIMENTAL RESULTS

In this section, we present the numerical experiments conducted using various open-source large language models (LLMs) applied to hospital discharge reports sourced from the MIMIC-IV dataset. Now we detail the experimental setup and discuss the outcomes derived from these evaluations.

### A. Open-source LLMs and Dataset

In this study, we utilize open-source LLMs to summarize hospital discharge reports. We selected a diverse set of LLMs for evaluation, as discussed in Sec. I. Table I provides an overview of these models along with their respective parameter counts. Here, we employ a subset of 100 hospital discharge reports sourced from the publicly available MIMIC-IV dataset, consistent with those used in [32]. Each discharge report is sequentially processed using each LLM's standardized prompt.

> You are a helpful assistant that helps patients understand their medical records. You will be given some doctor's notes, and you will need to summarize the patient's brief hospital course in one paragraph (within 1000 characters). Please only include important and essential information and avoid using medical jargon, and you MUST start the summary with "You were admitted".

### B. Key Information Extraction

We assess open-source LLMs on extracting (i) hospital admission reasons, (ii) key events occurred during the hospital stay, and (iii) follow-ups, from the hospital discharge reports, using GPT-4 as an evaluator [31]. The key information extraction step uses the following prompt:

> From the provided hospital discharge report, identify (i) three primary reasons for the patient's admission, (ii) three most important events that occurred during the hospital stay, and (iii) three crucial follow-up actions required after the discharge from the hospital.

Next, we evaluate whether these extracted elements are reflected in the LLM-generated summary. The following prompt is used:

> Given the extracted hospital information: admission reasons, hospital events and follow-up actions; and the provided model-generated summary: "the corresponding summary", identify how many of the reasons, important events, and follow-up actions are fairly covered in the summary.

Note that we acknowledge the potential risk of evaluator bias when using GPT-4 [35], however, we selected GPT-4 due to its high agreement with domain experts in prior studies on medical summarization. We also manually reviewed a subset of samples to calibrate model judgments and found strong alignment with human expectations. Figure 4 shows an original hospital discharge report and the corresponding LLaMA-generated summary. In this example, GPT-4 correctly identifies that, while LLaMA captures the admission reasons well, it falls short in covering the follow-up actions. Next, Table I shows the percentage of key details captured, highlighting each model's strengths and limitations in medical text understanding. The corresponding outputs from open-source LLMs and the judge model (GPT) are available in [36].

From the results, Qwen2.5 and DeepSeek-v2 emerge as the top-performing models in extracting admission reasons, with Qwen2.5 achieving $83.33\%$ comprehensively and $85\%$ fairly. Similarly, Phi3 and DeepSeek-v2 show strong performance in capturing hospitalization events, while LLaMA3.1 and LLaVA exhibit relatively lower effectiveness. Follow-up recommendations prove to be the most challenging category for all models, with comprehensive coverage ranging from $29.33\%$ (LLaMA3.1) to $55\%$ (Phi3). A possible explanation is that admission reasons are typically stated early in discharge reports, while follow-up instructions are scattered across different sections, making them harder to capture. However, fair coverage is comparatively better, with Phi3 and LLaVA achieving above $57\%$. These indicate that while LLMs perform reasonably well in extracting reasons for admission and key hospitalization events, their effectiveness in summarizing necessary follow-ups remains limited and requires further enhancement.

The observed limitations in extracting key events cannot be solely attributed to the 1000-character summary limit. As shown in Table I, many summaries fall well below this threshold yet still miss critical details like admission reasons, major events, and follow-ups. This suggests that the issue lies more in how the models prioritize information rather than in the imposed length constraint. Interestingly, some models, such as Phi3 and MistralLite, exceeded the given character limit, indicating that the restriction was not strictly enforced across all models. Their performance could improve with fine-tuning to better use available space and ensure comprehensive event extraction.

### C. Hallucinations

Now, we examine unsupported facts and incorrect or contradicted fact hallucinations in LLM-generated hospital discharge summaries, which can impact their reliability. Unsupported fact hallucinations introduce details absent in the original report, while incorrect/contradicted fact hallucinations conflict with the source text. Table II summarizes the hallucinations identified across different LLMs from 100 discharge report summaries obtained from each of the LLMs.

The results reveal notable variations in hallucination tendencies among models. Phi3 exhibits the highest number of hallucinations in both categories, with 150 unsupported fact hallucinations and 111 incorrect/contradicted fact hallucina-

TABLE I: Comparison of LLMs in terms of the total number of extracted key events from the respective summaries. While LLMs perform quite well (up to 83.33%) to find out the reasons, their performance is limited to figuring out the follow-ups (can be as low as 29.33%).

| LLM | # Parameters | Avg. Summary Length (SD) | "Comprehensively" Covered | | | "Fairly" Covered | | |
|---|---|---|---|---|---|---|---|---|
| | | | Reasons | Events | Follow-up | Reasons | Events | Follow-up |
| LLaMA3.1 | 8B | 674 (95) | 70.33% | 58% | 29.33% | 71.67% | 59.67% | 30.33% |
| Qwen2.5 | 7B | 778 (160) | **83.33%** | **70%** | 50.67% | **85%** | **70.67%** | 52.33% |
| DeepSeek-v2 | 16B | 1010 (332) | 80.33% | 65% | 41.33% | 81% | 67.33% | 42.67% |
| Phi3 | 3.8B | 1695 (580) | 81.33% | 63% | **55%** | **83%** | 65.67% | **58.33%** |
| Gemma2 | 9B | 621 (124) | 69.67% | 55.67% | 30% | 70.67% | 58.33% | 31.33% |
| MistralLite | 7B | 1400 (795) | 77.67% | 69% | 49% | 78.67% | 70% | 50% |
| LLaVA | 7B | 993 (416) | 70.67% | 61.67% | 53.33% | 73.33% | 64.33% | 57.33% |

**Brief Hospital Course:** This is a . . . . clotted AV fistula: Patient came from . . . to canulate the AV graft on the left arm. The patient underwent thrombectomy . . . . End stage kidney disease on hemodyalisis: The patient is on 3x/weeks dialysis schedule. . . . Pancytopenia: unclear cause at this point. The pt's platelets on admission were 67. . . . The cause of the pancytopenia was unclear at the time of discharge and further investigation of this problem was deferred to the outpatient care provider . . . Given the history of encephalopathy complicating his HCV infection, HCV treatment is highly recommended . . . HCV treatment with the patient deferred to the outpatient care provider. Transitional issues: the patient is noted to have pancytopenia . . . . Would defer for the PCP for further investigation. The patient has HCV infection with history of encephalopathy. If the encephalopathy was due to HCV infection, it would be a strong indication for starting HCV treatment. The patient has had a thrombectomy as treatment of thrombosed HD graft on the left side. Would recommend regular graft check and care. The graft should undergo regular maintenance every 3 months. . .

**Summary by LLaMA:** You were admitted to the hospital due to a clotted AV (arteriovenous) fistula in your left arm, which was preventing dialysis treatments for kidney failure. Your doctors performed a successful thrombectomy (a procedure to clear the blockage) and you underwent dialysis afterwards without any complications. We also investigated some low blood cell counts, known as pancytopenia, but couldn't determine the cause during your hospital stay. We will follow up on this with your primary care doctor when you are discharged. Your hepatitis C infection is a chronic condition that we recommended further treatment for, especially since it may have contributed to a previous episode of encephalopathy (brain damage).

Fig. 4: Comparison of the original hospital course note and LLaMA-generated summary: while LLaMA summary captures the primary admission reasons (including thrombosed AV fistula and kidney failure) and the hospital events (including thrombectomy and dialysis), it omits key follow-up plans: outpatient discussion for HCV treatment and routine AV graft maintenance every three months.

TABLE II: Comparison of LLMs in terms of total number of hallucinations within the considered 100 summaries.

| LLM | Total number of Unsupported fact Hallucinations | Total number of Incorrect/Contradicted fact Hallucinations |
|---|---|---|
| LLaMA3.1 | 85 | **27** |
| Qwen2.5 | 92 | 30 |
| DeepSeek-v2 | **46** | 102 |
| Phi3 | 150 | 111 |
| Gemma2 | 75 | **25** |
| MistralLite | 86 | 43 |
| LLaVA | 85 | 37 |

TABLE III: Comparison of LLMs in terms of the total number of extracted key events from the respective summaries using Gemini as the evaluator. Note that we omit Gemma2 from this comparison, as both Gemini and Gemma2 are developed by the same organization, which may introduce additional bias in the evaluation.

| LLM | Comprehensively Covered | | |
|---|---|---|---|
| | Reasons | Events | Follow up |
| LLaMA3.1 | 82.00% | 69.00% | 32.33% |
| Qwen2.5 | 86.67% | 77.00% | 52.33% |
| DeepSeekv2 | 91.00% | 70.33% | 44.67% |
| Phi3 | 81.67% | 69.33% | 57.00% |
| MistralLite | 75.00% | 64.67% | 44.67% |
| LLaVA | 76.67% | 63.00% | 48.67% |

tions, indicating significant reliability concerns. DeepSeek-v2, while generating fewer unsupported facts (46), shows a high number of contradicted facts (102), suggesting issues in accurately interpreting medical information. Other models, such as LLaMA3.1, Qwen2.5, and Gemma2, show more balanced yet still concerning levels of hallucinations, while MistralLite and LLaVA tend to generate a higher number of unsupported facts. These findings underscore the need for better fine-tuning and fact-verification, positioning this work as a crucial diagnostic step toward safer clinical NLP deployment.

*D. Evaluator Robustness: Extending to Gemini*

While GPT-4 has been used as the primary evaluator throughout this paper, we now assess the reliability of our evaluation framework by incorporating Gemini as an alternative evaluator. Table III presents the results based on Gemini's assessments. We find that open-source LLMs identify admission

reasons quite well with very high accuracy, but often struggle with follow-up actions, with coverage dropping as low as 32%. Despite minor variations in scores, the trends and model rankings remain largely consistent with those obtained using GPT-4, suggesting that our findings are robust across strong evaluators. These similarities suggest that our findings are not heavily dependent on the choice of evaluator, reinforcing the validity of our conclusions regarding LLM performance in clinical summarization tasks.

## V. CONCLUSION

Our study of open-source large language models (LLMs) for summarizing hospital discharge reports reveals their potential to accurately extract crucial clinical information but also highlights their susceptibility to generating hallucinations, including unsupported and incorrect facts. These inaccuracies

pose risks to patient care and safety. Our simulations show performance variability: while models (e.g. Qwen2.5 and DeepSeek-v2) capture admission reasons and events well, they often miss follow-up recommendations, highlighting the need for refinement and better medical understanding.

Future work can focus on developing automatic hallucination detection techniques [37] to identify and flag inaccurate or unsupported content in clinical text summaries. Fine-tuning models to enhance summary quality and reduce hallucinations can be another key direction. Additionally, integrating robust validation methods into clinical workflows can improve the reliability of LLM-generated summaries, offering healthcare professionals more efficient and accurate tools for patient care. Evaluating these summaries using readability metrics (e.g., Flesch-Kincaid) can further help assess their clarity and usefulness for patients. The analysis could be extended beyond discharge summaries to include other clinical documents such as progress and radiology reports. Furthermore, one can explore domain-specific hallucination clustering and temporal reasoning challenges in extracting follow-up instructions, aiming to guide safer real-world deployment of clinical NLP systems.

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
