# OpenReview forum: "Hallucinations and Key Information Extraction in Medical Texts: A Comprehensive Assessment of Open-Source Large Language Models"
_IEEE.org/EMBS/BHI/2025/Conference — BHI 2025_

### Official Review · Reviewer_VrnY · 2025-06-27
**Hallucinations and Key Information Extraction  in Medical Texts: A Comprehensive Assessment of  Open-Source Large Language Models**

**Confidence:** 5
**Clarity Of Writing:** great
**Clinical Significance:** good
**Methodological Novelty:** fair
**Overall Rating:** 5

**Experiments And Results:**

fair

**Questions For The Authors:**

1. Based on the evaluation criteria, why choose GPT-4 as a sole evaluator? Any proof that GPT-4 is the only evaluator?
2. Some errors seem to stem from misinterpreting the temporal sequence eg. “biopsy is recommended” vs. “biopsy was already rescheduled”). How did you control for this?
3. To help understand, was LLMs the best open-source model available or what was the criteria to the selection?
4. How reproducible are hallucination counts across multiple runs with the same input? and is it clinically adopted?
5. Can the hallucination types, be linked to specific types of content , for examples the medications vs. diagnoses or follow-up?
6.Did you consider whether LLMs were penalized for simplifying medical jargon (a desirable trait) that may have been mistaken as omissions or hallucinations?

**Strengths:**

Paper had the following strengths:
- The paper tackles both factual extraction and error characterization, which is often missing in LLM evaluation
- The Evaluation is focused on high-stakes domain, and benchmarked on wide state of the art open-source software's and performing multi-model comparison.
- Classifying the hallucinations with fine distinction is clear, thus provides better evaluation metrix
- Outlining Follow-up Deficiencies, brings attention to the particularly poor coverage

**Summary Of The Paper:**

The paper evaluates effectiveness and limitations of open-source large language models in extracting medical text summaries. It focus on addressing accuracy, identification and classification of hallucinations in summaries. The paper benchmarked using several open-source LLMs—including LLaMA, Qwen2.5, DeepSeek, Phi3, MistralLite, and others—against criteria for extracting essential clinical details, follow-up recommendations and minimizing hallucinated content. All the evaluation were conducted and standardized using GTP-4 as judgment model. On findings, model performed reasonable well and underperformed on follow-up. ANd in conclusion, it pave way for need to fine tune validation techniques to enhance reliablity in clinical applications.

**Weaknesses:**

- Reliance on GPT-4 as Sole Judge, this may introduce evaluator bias or risk or reinforce limitations from another black-box system especially when measuring credibility.
- The paper highlights risks, but it does not quantify the clinical severity of different hallucination types, such that are some more dangerous than others?
- No demonstration of any exploratory steps toward these solutions (e.g., even with one model) usedin fine-tuning and hallucination mitigation.
- The 1000-character limit isn’t strictly enforced, introducing inconsistency and possibly skewing comparative results
- It lacks human comparison apart from the notes , no benchmark given to support the LLM perfomance more meaningful.

---

### Official Review · Reviewer_aYV1 · 2025-07-02
**Use of LLMs for Clinical Summarization of Discharge Summaries for Patient Use**

**Confidence:** 4
**Clarity Of Writing:** good
**Clinical Significance:** good
**Methodological Novelty:** fair
**Overall Rating:** 6
**Final Rating:** 7

**Experiments And Results:**

good

**Questions For The Authors:**

Can you provide more methodological detail on how discharge reports were selected, how hallucinations were labeled, and how GPT-4 was used to evaluate model outputs?

Have you considered adding a readability assessment (e.g., Flesch-Kincaid) to support claims about the patient-friendliness of the summaries?

Could you briefly discuss how you envision integrating these summaries into real clinical workflows, and what steps might be needed to validate their usability with patients?

**Strengths:**

The topic is highly relevant, addressing the timely issue of using LLMs for clinical summarization to aid patient understanding.

The paper provides useful comparative data across several open-source LLMs, rather than focusing only on proprietary systems.

The inclusion of concrete examples of discharge summaries and hallucination types helps illustrate model strengths and failure modes.

The structure is clear, and the tables effectively communicate key performance metrics.

**Summary Of The Paper:**

The paper evaluates the performance of several open-source large language models (LLMs) in generating patient-friendly discharge summaries from hospital discharge reports in the MIMIC-IV dataset. The study compares these models on two main tasks: (1) key information extraction (admission reasons, major hospital events, follow-up actions) and (2) hallucination detection (unsupported or contradicted facts). The authors present comparative quantitative results and provide examples of typical model outputs, highlighting challenges and limitations.

**Weaknesses:**

The Methods section is thin and lacks sufficient detail for reproducibility. Critical information is missing about dataset selection, model configurations, hallucination labeling procedures, and evaluation protocols (e.g., GPT-4 judge setup).

The study’s use case is framed as patient-facing discharge summarization, but no usability, readability, or end-user acceptance data are provided. It is unclear whether the summaries are useful or preferred by patients.

No statistical testing is reported to determine whether differences in model performance or hallucination rates are significant.

The paper focuses primarily on accuracy and hallucination detection but does not address practical deployment or integration concerns (e.g., clinical workflow impact

The paper title is not representative of the work actually done--too late to change--but if a full paper is submitted the title should be more descriptive of the actual work done.

---

### Official Review · Reviewer_SotM · 2025-07-13
**Summary of Submission Number: 70**

**Confidence:** 5
**Clarity Of Writing:** fair
**Clinical Significance:** fair
**Methodological Novelty:** poor
**Overall Rating:** 3
**Final Rating:** 5

**Experiments And Results:**

fair

**Questions For The Authors:**

1. Can you please explain why the provided prompt was used and not any other variation of this prompt?
2. Also can make sure to include important results in your abstract? It currently lacks any mention of the experimental results or performance metrics. Including key results or quantitative highlights would provide readers with a clearer summary of the study’s contributions and findings.
3. Overall, I'd highly recommend authors to see the above comments and improve the paper. Also, make sure that your future submissions are completely blind.

**Strengths:**

1. The authors did not exceed the page length limit
2. The topic is of importance as summarization leads to quick understanding for the physicians and can reduce the burden on healthcare systems.

**Summary Of The Paper:**

This study utilizes LLaMA, Mistral, Gemma, Phi, Falcon, LLaVA, DeepSeek, and Qwen for extracting key events from MIMIC-IV discharge reports (example: admission reasons, major in-hospital events, and critical follow-up actions). They also assess the prevalence of various types of hallucinations in the summaries produced by these models. The authors used GPT-4 as the evaluator.

**Weaknesses:**

1. It is my understanding that this manuscript violates the double-blind review policy by referencing the authors' GitHub repository: “The corresponding outputs from open-source LLMs and the judge model (GPT) are available in [34].” Such references should be removed or anonymized until after the review process. I leave it up to the AC for any decision in this regards.

2. The manuscript includes vague phrasing such as “The selected LLMs include LLaMA, DeepSeek, and Phi etc.”. In a scientific context, especially when comparing model performance, the list of models used should be stated in full. The use of "etc." is inappropriate and detracts from the clarity and reproducibility of the study. Also since this information was already presented before, I'd have preferred if the authors directly discussed results in the "Experimental results" seciton rather than repeating the same information.

3. While the authors do provide the prompt used for the experiments, they do not discuss how this prompt was developed. It would strengthen the work to explain the rationale behind the prompt design, and whether any prompt engineering or iterative refinement was applied. This is important, as prompt formulation can significantly affect model output quality.

4. The paper mentions that GPT-4 is used as an evaluator for model outputs, but does not detail how GPT-4 was prompted to perform this evaluation. This information is crucial for reproducibility and for assessing the validity of the evaluation results.

5. The manuscript contains repetitive information, particularly in restating the study’s goals and methods multiple times. For example, by the time the reader reaches the “Experimental Results” section, the purpose of utilizing open-source LLMs for summarizing discharge reports has already been reiterated several times. Streamlining this content and focusing on presenting results would improve clarity and readability.

6. Figure 1 is not cited or discussed anywhere in the main text. Additionally, its relevance and meaning within the context of the study are unclear. Every figure should be properly referenced and its significance explained in the text.

7. There is no dedicated section/paragraph regarding dataset, making it difficult to determine the scope of the data used. While the authors mention using the MIMIC-IV dataset (which a reader can perhaps read more about), they do not specify the number of discharge summaries analyzed. It is only in Section IV.C (“Hallucinations”) that the reader learns that 100 summaries were used, and even then, this is only in the context of hallucination analysis. A clear, early explanation of dataset size and selection criteria is necessary.

---

### Official Review · Reviewer_tpKJ · 2025-07-17
**A Comprehensive Study on Open-Source LLMs for Clinical Text Summarization**

**Confidence:** 3
**Clarity Of Writing:** excellent
**Clinical Significance:** great
**Methodological Novelty:** fair
**Overall Rating:** 7
**Final Rating:** 7

**Experiments And Results:**

good

**Questions For The Authors:**

Although the Falcon LLM is used in your research, why do Tables I and II not show its parameters and performance?

**Strengths:**

1) The paper tackles a problem of high significance for the health informatics community. Automating clinical summarization can greatly enhance workflow efficiency and patient understanding, but only if the summaries are reliable. The focus on key information extraction and hallucination detection directly addresses the primary barriers to the safe deployment of LLMs in clinical practice.
2) The study is well-structured and methodologically sound. The authors assess a wide range of current, open-source LLMs, which is a valuable contribution to the field, as much research focuses only on proprietary models.

**Summary Of The Paper:**

This paper presents a timely and relevant study on the application of open-source LLMs for summarizing clinical discharge reports. The authors conduct a comprehensive evaluation of several prominent open-source LLMs, including models from the LLaMA, Qwen, and Phi families, using the MIMIC-IV dataset. The core of the research focuses on two critical challenges: the models' ability to accurately extract key clinical information and their propensity to generate various types of "hallucinations". The study finds that while some models like Qwen2.5 and DeepSeek-v2 perform well in identifying admission reasons, all models struggle with extracting crucial follow-up instructions.

**Weaknesses:**

1) While the evaluation is comprehensive, the paper doesn't introduce significantly new methodologies or novel approaches to summarization or hallucination detection.
2) Relying solely on GPT-4 as a judge model for assessing LLM-generated summaries may introduce potential biases in the evaluation process. An additional human expert review or an independent validation method could strengthen the evaluation.

---

### Official Review · Reviewer_peSm · 2025-07-18
**Evaluation of open-source LLMs for summarizing clinical discharge reports**

**Confidence:** 4
**Clarity Of Writing:** great
**Clinical Significance:** great
**Methodological Novelty:** good
**Overall Rating:** 7
**Final Rating:** 7

**Experiments And Results:**

great

**Questions For The Authors:**

Were any human evaluations conducted to calibrate or confirm GPT-4 judgments? How consistent were they if that were the case?

Are hallucinations concentrated in certain clinical domains?

**Strengths:**

working on relevant interdisciplinary problem, and have detailed taxonomy and examples of the different hallucination types, evaluated across multiple models, have statistical results to support claims

**Summary Of The Paper:**

This paper evaluates the performance of several open-source large language models (LLMs) in extracting key information and minimizing hallucinations in medical text summarization. The paper uses discharge summaries from the MIMIC-IV dataset, and then the authors assess how well models capture clinical events such as admission reasons and follow-ups, and quantify hallucinations into types like unsupported or contradicted facts. GPT-4 serves as an evaluator of model output.

**Weaknesses:**

It was mentioned that GPT-4 was used as an evaluator. Have there been thoughts on how this reliance may introduce bias?

Would appreciate more discussion on feasibility for real-world deployment.

Follow-up extraction performance was notably low across models. This could be developed more.

Other than discharge summaries, could you consider extending to other clinical documents

I would try to move the tables on page 6 to be before the references section starts